# Improving Cognitive Function with Nutritional Supplements in Aging: A Comprehensive Narrative Review of Clinical Studies Investigating the Effects of Vitamins, Minerals, Antioxidants, and Other Dietary Supplements

**DOI:** 10.3390/nu15245116

**Published:** 2023-12-15

**Authors:** Mónika Fekete, Andrea Lehoczki, Stefano Tarantini, Vince Fazekas-Pongor, Tamás Csípő, Zoltán Csizmadia, János Tamás Varga

**Affiliations:** 1Department of Public Health, Faculty of Medicine, Semmelweis University, 1089 Budapest, Hungary; fekete.monika@med.semmelweis-univ.hu (M.F.); stefano-tarantini@ouhsc.edu (S.T.);; 2National Institute for Haematology and Infectious Diseases, Department of Haematology and Stem Cell Transplantation, South Pest Central Hospital, 1097 Budapest, Hungary; ceglediandi@freemail.hu; 3Department of Neurosurgery, The University of Oklahoma Health Sciences Center, Oklahoma City, OK 73104, USA; 4Department of Health Promotion Sciences, College of Public Health, The University of Oklahoma Health Sciences Center, Oklahoma City, OK 73104, USA; 5Peggy and Charles Stephenson Oklahoma Cancer Center, Oklahoma City, OK 73104, USA; 6Faculty of Health Sciences, University of Pécs, 7621 Pécs, Hungary; penituki@gmail.com; 7Department of Pulmonology, Semmelweis University, 1083 Budapest, Hungary

**Keywords:** dementia, cognitive function, dietary supplement intervention, randomized controlled trial, vitamin, mineral, antioxidant, omega-3 polyunsaturated fatty acid

## Abstract

Cognitive impairment and dementia are burgeoning public health concerns, especially given the increasing longevity of the global population. These conditions not only affect the quality of life of individuals and their families, but also pose significant economic burdens on healthcare systems. In this context, our comprehensive narrative review critically examines the role of nutritional supplements in mitigating cognitive decline. Amidst growing interest in non-pharmacological interventions for cognitive enhancement, this review delves into the efficacy of vitamins, minerals, antioxidants, and other dietary supplements. Through a systematic evaluation of randomized controlled trials, observational studies, and meta-analysis, this review focuses on outcomes such as memory enhancement, attention improvement, executive function support, and neuroprotection. The findings suggest a complex interplay between nutritional supplementation and cognitive health, with some supplements showing promising results and others displaying limited or context-dependent effectiveness. The review highlights the importance of dosage, bioavailability, and individual differences in response to supplementation. Additionally, it addresses safety concerns and potential interactions with conventional treatments. By providing a clear overview of current scientific knowledge, this review aims to guide healthcare professionals and researchers in making informed decisions about the use of nutritional supplements for cognitive health.

## 1. Introduction

Cognitive impairment and dementia are among the most significant health challenges of our time, especially as the global population ages [1,2,3,4]. The prevalence of dementia increases exponentially with advancing age [5,6,7], with a prevalence of 0.8% to 6.4% in the population over 65 years of age [8], and 28.5% at age 90 in the European Union [9]. The World Health Organization (WHO) estimates that approximately 50 million people worldwide live with dementia, a number expected to triple by 2050 [10]. Dementia, a syndrome encompassing over 200 conditions [11], is characterized by progressive cognitive impairment [12,13,14] and a decline in functional abilities, often accompanied by behavioral and psychological symptoms [15]. Two of the most prevalent forms of dementia are Vascular Cognitive Impairment (VCI) and Alzheimer’s Disease (AD), both contributing significantly to the global dementia burden [16,17,18,19]. These conditions not only impact the individuals suffering from them, but also place a considerable strain on families, caregivers, and healthcare systems. The societal and economic implications are profound, encompassing lost productivity, increased healthcare costs, and substantial emotional and physical burdens on caregivers [15].

The search for effective interventions to prevent, delay, or ameliorate cognitive decline is of paramount importance, especially given the complex mechanisms underlying age-related cognitive decline and dementia [20]. These mechanisms encompass a spectrum of pathologies, ranging from microvascular issues [21], including blood–brain barrier (BBB) disruption [22,23,24,25], impaired cerebral blood flow regulation [26,27,28,29], impaired glymphatic function [30], and small vessel disease [31,32] to macrovascular pathologies such as atherosclerosis [33] and stroke. Additionally, neuroinflammation [34,35], synapse loss, white matter damage [36,37] and changes in connectivity [38,39], neuronal metabolic dysfunction [40,41] and amyloid pathologies [42,43] play significant roles in the progression of cognitive impairment and dementia. These multifaceted and interrelated pathologies highlight the complexity of brain aging and the challenges in mitigating cognitive decline. While pharmacological treatments have shown some benefits, they may not fully address the multifaceted nature of cognitive decline [44,45,46]. This has led to growing interest in alternative approaches [47,48,49,50], particularly dietary interventions and nutritional supplementation [51,52,53,54], as a potential means to support cognitive health. Nutrition plays a crucial role in brain health, and deficiencies in specific nutrients have been linked to cognitive impairments [55,56,57]. This review aims to synthesize the current clinical research on the efficacy of various nutritional supplements—including vitamins, minerals, antioxidants, and other dietary components—in enhancing cognitive function and potentially mitigating the risk or progression of dementia.

Recent studies, including findings from the Chicago Health and Aging Project (CHAP), have highlighted the importance of a healthy lifestyle, comprising a balanced diet, regular physical activity, and cognitive engagement, in extending life expectancy and delaying the onset of dementia [51,58]. Moreover, as the efficiency of nutrient absorption decreases with age and is affected by certain medications, the role of dietary supplementation, such as vitamins B, C, D, antioxidants, and polyunsaturated fatty acids (PUFAs), becomes increasingly significant [59,60,61]. Moreover, elderly individuals typically engage in less physical activity and spend reduced time outdoors, leading to lower levels of sun exposure [62] and, consequently, diminished vitamin D synthesis in their skin. Given these factors, vitamin D supplementation emerges as a crucial intervention for health protection in the elderly [63]. The typical diet often lacks sufficient omega-3 PUFAs, leading to recommendations for consuming fatty fish like tuna or salmon twice a week, or taking fish oil capsules, particularly for those on a vegan diet [52,64,65,66,67,68,69,70,71,72].

This review critically examines the potential of nutritional supplements in addressing cognitive decline and dementia, taking into consideration the complexity of recommending dietary supplements based on individual dietary habits and health conditions. We aim to provide a comprehensive overview of the evidence supporting the use of nutritional supplements for cognitive health and to inform healthcare professionals and researchers about their potential role in a comprehensive approach to brain health.

## 2. Methods

We conducted a literature search on dietary supplement interventions in the Cochrane Central Register of Controlled Trials (CENTRAL) databases, PubMed, and ClinicalTrials.gov from 31 October 2018 to 31 October 2023. We focused exclusively on randomized controlled trials (RCTs) and human clinical trials, seeking correlations with cognitive function. The search used specific and MESH keywords, including vitamins: “Vitamin A”, “Vitamin B 6”, “Vitamin B 9”, “Vitamin B 12”, “Vitamin C”, “Vitamin D”, “vitamin E”, “minerals”, “antioxidants”, “flavonoids”, “carotenoids”, “omega 3 fatty acids”, “cognition”, “memory”, “executive function”, “dementia”, “Alzheimer disease”, “cognition disorders”, “randomized controlled trial”, “controlled clinical trial”, without language restrictions. Between the words of the search words listed, we used the conjunctions “AND” or “OR”. Indexed duplicate articles were removed, then titles and abstracts were screened; those that did not meet inclusion criteria were excluded. The articles finally selected were carefully evaluated on the basis of their full text. This review does not cover the application of complex diets, such as the Dietary Approach to Stop Hypertension (DASH diet), high-protein diet, ketogenic diet, low-fat diet, the Mediterranean diet, low glycaemic index (GI) Diet, Mediterranean-DASH Intervention for Neurodegenerative Delay (MIND), nor does it include animal experiments or in vitro research. The role of probiotics, prebiotics, and symbiotics in preserving and enhancing healthy cognitive functions has been demonstrated; however, this summary does not encompass that area due to the limitations of the article’s length. The aim of this review was to provide an up-to-date overview of recent findings on the relationship between the use of different dietary supplements (e.g., vitamins, minerals, antioxidants, essential fatty acids) and cognitive function, according to the PICO (Population, Intervention, Comparison, and Outcomes) criteria. The inclusion and exclusion criteria for the studies included in this review are given in Table 1. Figure 1 shows the flowchart of the articles involved in the selection process. In total, 42 articles and 11,913 patients were included in this review.

## 3. Results

### 3.1. Vitamin A Supplementation

During the study period, no databases included randomized clinical controlled trials solely focusing on Vitamin A supplementation. However, its combined use with other supplements is discussed later in this article.

### 3.2. Vitamin B Supplementation and Cognitive Function

With regard to B vitamins, previous research has elucidated the beneficial role of folic acid, B12, and B6 vitamins in preserving cognitive functions. In the present study, we observed that when vitamin B supplementation was administered independently to patients [73,74] and when it was administered concomitantly with magnesium, vitamin D, vitamin C [75], and docosahexaenoic acid (DHA) [76], resulted in significant enhancements and substantial benefits in terms of memory and cognitive augmentation among healthy adults (*p* < 0.001), particularly when magnesium and antioxidant vitamins were included [75]. The combined supplementation with folic acid and vitamin B12 resulted in statistically significant improvements in cognitive scores, as reported by Ma F et al. [77]. In a study [76], folic acid + DHA supplementation significantly increased measured cognitive scores in comparison to the placebo group. The study by Ma F et al. [74] revealed that a 24-month regimen of folic acid supplementation (400 µg/day) significantly improved cognitive function and reduced blood biomarker levels of amyloid β protein (Aβ) in mild cognitive impairment (*p* < 0.05). In a study [78], the daily oral administration of folic acid (FA), DHA, and their concomitant administration for six months resulted in a significant enhancement in the Full Scale Intelligence Quotient (FSIQ) and specific subtests of the Wechsler Adult Intelligence Scale in comparison to the placebo group (*p* < 0.05). A summary of clinical investigations on the correlation between vitamin B supplementation and cognitive functions is presented in Table 2.

### 3.3. Results on Antioxidants and Cognitive Function

Prominent antioxidants include specific vitamins, namely vitamins A, C, and E, minerals such as selenium, chromium, and zinc, as well as carotenoids, flavonoids, polyphenols, and lignans. These components protect against free radicals and help maintain healthy functioning of cognitive processes, such as memory, cognition, and concentration. In our current summary research, in a study by Thaung Zaw JJ et al. [80], a daily supplementation of 75 mg resveratrol resulted in a significant improvement in overall cognitive performance among middle-aged women. Foroumandi E et al. [81] demonstrated promising positive effects on memory, quality of life, and levels of selective oxidative indices in patients with mild to moderate Alzheimer’s disease (AD) who received 5 cc of fenugreek seed extract (equivalent to 500 mg of dry extract) (*p* < 0.001) for 4 months. The supplementation of 1000 mg/day of vitamin C [82], carotenoids: lutein and zeaxanthin (13 mg, or 27 mg/day) supplementation [83], as well as 1.6 g/day of alpha-lipoic acid [84] resulted in significant improvement compared to the control group in terms of attention, memory, and other observed cognitive tests (*p* < 0.05 for all). Additionally, administering 3 g of desert olive tree pearls/day for three months significantly improved (*p* < 0.05) psychomotor speed, reaction time, cognitive flexibility, processing speed, and executive function domains [85]. However, no significant improvement in cognitive tests was observed in healthy individuals supplemented with cocoa extract (containing 500 mg/day of flavanols) or Ginkgo biloba (240 mg/day in patients with Alzheimer’s disease) (Table 3).

### 3.4. Vitamin D Supplementation and Cognitive Function

Maintaining optimal levels of vitamin D can be effectively achieved through proper nutrition and time spent outdoors during the spring and summer months when sunlight exposure is higher. However, in autumn and spring, when the number of sunshine hours is reduced, it is advisable to supplement with vitamin D. Expert recommendations suggest that healthy adults should consume 1500–2000 IU of vitamin D daily, and even doubling this value to 4000 IU daily possible in some cases. In our review, we have observed that vitamin D is of utmost importance for mood and cognitive functions. Vitamin D supplementation may play a role in preventing dementia and enhancing cognitive function, although a study [90] involving 600 IU of vitamin D3 per day showed no significant effects on cognitive functions or mood. Similarly, Jorde R et al. [91] and Bischoff-Ferrari HA et al. [92] observed no substantial improvement in cognitive function with vitamin D supplementation.

Contrary to these studies, Jia J et al. [93], Yang T et al. [94], Castle M et al. [95], Ghaderi A et al. [96] and Hu J et al. [97] reported significant enhancements in cognitive performance as a result of vitamin D supplementation. A study [98] did not support a significant cognitive benefit from 2000 IU per day compared to 800 IU per day among healthy older adults. Meanwhile, Castle M et al. [95] described differential effects of vitamin D3 supplementation (600, 2000, or 4000 IU per day) on cognitive measurements across various functional domains (Table 4).

### 3.5. Omega-3 Dietary Supplements and Cognitive Function

Consistent use of omega-3 dietary supplements may be correlated with enhancements in cognitive ability and reduced susceptibility to cognitive deterioration. In addition to their pivotal role in immune function, omega-3 polyunsaturated fatty acids have the potential to improve memory and reduce the risk of chronic degenerative diseases such as Alzheimer’s disease and attention disorders. Empirical studies have revealed that omega-3 fatty acids may help improve memory and cognitive functions in various forms of dementia and reduce the risk of these conditions. It has also been documented that omega-3 fatty acids help maintain brain health and cognitive function in healthy adults. In our study, we found that patients who received 1 g of fish oil per day, either alone or in combination with vitamin E and carotenoids, experienced memory and mood improvement, a significant reduction in the number of errors in memory tasks, and enhancements in attention and executive functions [103,104,105,106]. Rasmussen J et al. [107] have given patients in the early stages of Alzheimer’s disease a safe and well-tolerated combination of docosahexaenoic acid, eicosapentaenoic acid, uridine monophosphate, choline, phospholipids, selenium, folic acid, and vitamins B12 and E, and have reported clinically significant benefits in various cognitive functions (e.g., attention, memory, executive function, *p* < 0.05). Nevertheless, we have encountered studies in which fish oil supplementation did not influence cognitive performance, memory, or mood, and the final analysis did not yield any evidence of its impact on cognitive function measurement [108,109,110,111,112] (Table 5).

## 4. Discussion

In our current comprehensive research overview, we highlight the indisputable significance of various vitamins, minerals, antioxidants, and essential fatty acids in enhancing cognitive function. These essential nutrients play a pivotal role in sustaining the proper functioning of the central nervous system. As individuals age, nutrient deficiencies become more common, stemming from inadequate diet, reduced nutrient absorption, or medication-related side effects. Supplementation of these elements is, therefore, essential not only for the proper functioning of the immune system but also for the preservation of cognitive abilities, memory, and cognitive acumen. B vitamins are well documented to have profound effects on cognitive function, memory retention, and concentration. Deficiencies can lead to memory impairment and concentration deficits, and supplementation can improve cognitive abilities. Free radicals that impact neurons can cause damage or even complete loss, thereby contributing to the development of neurodegenerative diseases. Our comprehensive review underscores the critical importance of nutritional supplements, including various vitamins, minerals, antioxidants, and essential fatty acids in preserving/enhancing cognitive function in older adults. These supplements are pivotal in maintaining the central nervous system’s proper functioning, especially as nutrient deficiencies become more prevalent with age due to factors like inadequate diets, reduced nutrient absorption, or medication side effects. Supplementing these elements is crucial for preserving cognitive abilities, including memory and executive function.

### 4.1. Vitamin B

B vitamins play a crucial role in maintaining cognitive function during aging, acting as essential cofactors in various neurological processes [115]. As individuals age, the risk of B vitamin deficiencies increases, potentially leading to cognitive decline [116]. Vitamins B6, B9 (folate), and B12 are particularly important, as they contribute to the maintenance of normal brain function by influencing homocysteine metabolism, an amino acid linked to neurodegenerative diseases and cardiovascular and cerebrovascular impairment when present at elevated levels [117,118]. Studies [119,120,121] have demonstrated that a high intake of these B vitamins can significantly slow cognitive decline in older adults, particularly in those with elevated homocysteine levels or mild cognitive impairment [116,122]. For instance, supplementation with B vitamins has been shown to lower homocysteine levels, thereby reducing the rate of brain atrophy and improving cognitive outcomes [123]. Additionally, B vitamins are involved in the synthesis of neurotransmitters and the maintenance of myelin, the protective sheath around nerve fibers, which is crucial for efficient brain signaling [123]. This protective effect of B vitamins against cognitive decline is especially beneficial for older adults who are at an increased risk of dementia, suggesting that adequate intake of these nutrients could play a key role in preserving cognitive health in the aging population [115,116,119,120,122].

### 4.2. Vitamin C, Vitamin E and Other Antioxidants

Supplementation with vitamin C, vitamin E, and other antioxidants plays a potential important role in preserving cognitive function during aging, primarily through their ability to combat oxidative stress [124], a key factor in cellular aging [125] and the development of cardiovascular [126] and cerebrovascular impairment [127], neurodegeneration [128] and age-related cognitive decline [129].

Vitamin C is a potent antioxidant that can help protect the brain from oxidative stress, which can impair cognitive functions [130]. In addition, vitamin C plays a pivotal role in the synthesis and functioning of the neurotransmitters dopamine and noradrenaline found in the brain [130]. Several studies have explored the associations between vitamin C and physical and mental well-being; its deficiencies may increase the risk of severe conditions such as cancer, heart disease, and diabetes [131,132,133]. Research has also found links between vitamin C deficiency and attention, concentration, executive function, memory, linguistic and conceptual thinking [134,135,136]. Low levels of vitamin C can negatively affect mood, potentially leading to depression and cognitive impairment [137], in other words, higher levels of vitamin C are correlated with improved mood and reduced depression and confusion [134]. A meta-analysis has described how vitamin C supplementation can enhance the mood in individuals with depression [138]. Consequently, this can improve cognitive performance and reduce the “brain fog” associated with depression [138]. Further research and long-term follow-up studies involving more patients are needed to establish the preventive effect of vitamin C against the development and progression of Alzheimer’s disease [139]. In conclusion, there existing evidence suggests that maintaining healthy levels of vitamin C may be protective against age-related cognitive decline and neurodegenerative diseases, and that cognitive function improves with vitamin C supplementation [139,140].

Vitamin E is a potent fat-soluble antioxidant. It interacts and synergizes with several other antioxidants, such as glutathione, selenium, vitamin C, carotenes, and carotenoids [141]. Vitamin E protects brain cells from oxidative stress-induced damage [142] and is particularly effective in maintaining neuronal integrity and function. It is known for its protective effects against lipid peroxidation in cell membranes, which is vital for preserving cognitive health. Studies have shown that individuals with a higher intake or serum levels of these antioxidants have a lower risk of cognitive decline and dementia, including Alzheimer’s disease [141,142,143]. The existing evidence supports the inclusion of vitamin E supplementation in a dietary protocol aimed at preserving cognitive health in the elderly.

Polyphenols, including resveratrol and curcumin, represent a diverse group of compounds with potent antioxidant and anti-inflammatory properties, playing a significant role in the preservation of cognitive function in aging. Resveratrol, found in grapes and red wine, has gained attention for its potential neuroprotective effects. It is believed to activate pathways that help in protecting cells from damage and improve blood flow to the brain, thereby potentially enhancing cognitive functions and reducing the risk of neurodegenerative diseases [144]. Clinical evidence suggests that resveratrol supplementation can improve memory and cognitive performance in older adults [144,145].

Curcumin, the active component of turmeric, is another widely studied polyphenol known for its strong anti-inflammatory and antioxidant properties. It has been shown to cross the blood–brain barrier and exert neuroprotective effects, potentially helping in the prevention of age-related cognitive decline. Studies indicate that curcumin may improve memory and mood in people with mild, age-related memory loss [146,147].

Other polyphenols, like flavonoids, found in berries, tea, and cocoa, are also crucial for cognitive health [148,149]. Flavonoids have been associated with improved cognitive abilities, reduced risk of dementia, and enhanced memory and learning in older adults [150]. For example, epigallocatechin gallate (EGCG) from green tea has been studied for its role in protecting neurons, reducing the formation of amyloid plaques, and improving cognitive function [151].

Another class of polyphenols, anthocyanins, found in dark-colored fruits like blueberries, have been shown to improve neural signaling and enhance memory [152]. Regular consumption or supplementation with these polyphenols can provide antioxidative and anti-inflammatory benefits, which are particularly beneficial in countering age-related cognitive decline [145]. However, while the biological effects of these compounds are well-documented, more large-scale, long-term clinical trials are needed to conclusively establish their efficacy in preventing or slowing down cognitive impairment in older adults.

### 4.3. Vitamin D

Vitamin D, essential for overall health, plays a multifaceted role ranging from maintaining bone health to supporting immune system functions [153,154]. It aids in the absorption of calcium and phosphorus, crucial for optimal bone density, and is involved in normal muscle function and blood sugar metabolism [155,156]. Its significance extends to the development of the brain and nervous system in early childhood and is crucial for cognitive functions such as memory, thinking, and concentration in later life [157]. Vitamin D is also instrumental in producing neurotransmitters like dopamine and serotonin, which regulate mood and emotions [157].

Approximately 40% of the European population is affected by vitamin D deficiency, which can worsen during the late winter months [158]. Consequently, the recommended daily intake for adults has been increased to 2000 IU/day [158,159]. The elderly, along with individuals with chronic diseases, specific dietary preferences, or certain health conditions, are particularly vulnerable to this deficiency [160,161]. Notably, vitamin D deficiency is linked to an increased risk of diseases like type 2 diabetes, cancer, multiple sclerosis, depression, Parkinson’s, and Alzheimer’s disease [162,163]. A study [164] involving over 1600 elderly individuals revealed that those with mild vitamin D deficiency were over 50% more likely to develop dementia, and the risk was even higher in those with severely low levels. These findings underscore the importance of maintaining adequate vitamin D levels for cognitive health.

Clinical evidence supports the role of vitamin D supplementation in preserving cognitive function, especially in older adults [165,166,167]. Research has shown that adequate levels of vitamin D are associated with improved cognitive performance, and supplementation has been found to benefit those with deficiencies [165,166,167]. This is particularly crucial for cerebrovascular health, as vitamin D supports blood flow to the brain and reduces the risk of cerebrovascular diseases, which can impact cognitive abilities [162,163]. Its role in neuroprotection, neurotransmission, and brain plasticity highlights its potential as a key nutrient in maintaining brain health during aging [168,169]. Therefore, vitamin D supplementation could be a strategic approach in mitigating the risk of age-related cognitive impairment and supporting overall brain health in the elderly population. The exact mechanisms are thought to involve vitamin D’s role in neuroprotection, neurotransmission, and brain plasticity, highlighting its potential as a crucial nutrient in maintaining brain health with aging. However, while these associations are promising, further large-scale, long-term studies are needed to establish definitive causal links between vitamin D supplementation and cognitive function preservation in older adults [168,169].

### 4.4. Vitamin K

In the human body, vitamin K is an essential component for physiological processes and is also a fat-soluble vitamin [170]. It is naturally expressed in two forms, namely K1 (phylloquinone) and K2 (menaquinone), both of which play pivotal roles in a wide spectrum of physiological processes [170]. Its principal function is to modulate hemostasis through the synthesis of coagulation factors. In addition to these primary roles, vitamin K is also required for various functions in cell growth, proliferation, cell genesis, and apoptosis [170,171,172]. Maintenance of normal vitamin K levels may also contribute to the preservation of memory in the elderly [172]. There are studies suggesting a link between reduced serum concentrations of vitamin K and deterioration of cognitive function in the geriatric population (aged 65 years and above) [172,173,174]. Preclinical studies raise the possibility that vitamin K2 may protect nerve cells against the toxicity of amyloid β [175].

### 4.5. Omega-3 Polyunsaturated Fatty Acids

Omega-3 polyunsaturated fatty acids (PUFAs) [56,153] are increasingly recognized for their critical role in maintaining cognitive function during aging, with ongoing research continuously examining their positive effects on central nervous system functions, including memory, attention, concentration, learning capabilities, and overall well-being [52,65,176,177,178,179]. These fatty acids, particularly eicosapentaenoic acid (EPA) and docosahexaenoic acid (DHA), are crucial components of cell membranes in the brain and are involved in various neurophysiological processes [52,64,176,178,180,181,182,183]. DHA, the most abundant omega-3 fatty acid in the brain, is vital for the maintenance and function of neural cells [65,177,178,179]. It is known for its role in enhancing synaptic plasticity and neuronal signaling, which are key factors in cognitive processes like learning and memory [181]. EPA, on the other hand, contributes to cognitive health primarily through its anti-inflammatory properties, which are beneficial in reducing neuroinflammation, a contributing factor to cognitive decline and the development of neurodegenerative diseases [52,64,176,178,180,181,182,183,184,185]. Higher doses of omega-3, typically ranging from 500 to 2000 mg per day, have been suggested to alleviate symptoms of depression and anxiety, further supporting cognitive health [186]. The Western diet, unfortunately, tends to contain more omega-6 fatty acids compared to omega-3, which can be detrimental to health [187]. The optimal ratio of omega-6 to omega-3 is closer to 2:1, and since the body cannot produce these fatty acids, they must be obtained through diet or supplements, with effective doses for various health conditions ranging from 250 to 4000 mg of omega-3 per day [187,188]. Sources of omega-3 include fatty fish like salmon, mackerel, tuna, and plant sources such as flaxseed, chia seeds, nuts, and canola oil [52,64].

Regular consumption of omega-3 supplements is beneficial for cognitive and emotional development, reading skills, and cognitive and concentration abilities [52,65,176,177,178,179]. Furthermore, omega-3 PUFAs have shown potential in reducing the risk of neurodegenerative diseases like Alzheimer’s disease [189], potentially slowing the progression of the disease, delaying its onset, and being associated with improved cognitive abilities. Clinical studies have also demonstrated that regular intake of omega-3 fatty acids can alleviate symptoms of mental health disorders such as depression [176,190,191]. DHA is also essential for the retina, with deficiency leading to vision problems and increased risk of eye diseases [192]. Additionally, omega-3 fatty acids positively affect bones, muscles, and joints, enhancing bone strength and reducing the risk of osteoporosis [193,194]. Therefore, consistent use of omega-3 supplements has been associated with multifaceted health benefits, including improved cognitive ability and reduced susceptibility to cognitive decline [52,64,176,178,180,181,182,183].

### 4.6. Mineral Supplementation

Mineral supplementation plays a critical role in supporting brain and cognitive health, particularly in the aging population [195,196]. Minerals, being inorganic substances essential for the body’s physiological functions, cannot be synthesized endogenously and must be obtained through diet or supplementation. Their impact on cognitive abilities is increasingly recognized [195,196].

Magnesium supplementation, for instance, is associated with enhanced cognitive functions and a reduced risk of dementia [197]. Studies [197,198] have demonstrated that individuals with higher magnesium levels have a notably lower risk of dementia. Increased magnesium intake is particularly beneficial for brain health and may help in preserving intellectual function and reducing the risk of dementia, as it protects nerve cells and positively influences blood pressure.

Iron is vital for numerous physiological processes, including cognitive function and cellular metabolism [199,200,201,202,203,204]. Even mild forms of iron deficiency can impair concentration and reduce immunity, and are linked with delayed neurological development and poorer academic performance. Iron supplementation has been shown to positively affect intelligence quotient (IQ) scores and iron-deficiency anemia in the elderly is associated with an increased prevalence of dementia and Alzheimer’s disease [199,200,201,202,203,204].

Selenium is a crucial component of antioxidant enzymes like glutathione peroxidase [205]. These enzymes protect the brain from oxidative stress, which is implicated in aging and neurodegenerative diseases like Alzheimer’s and Parkinson’s [205]. Oxidative stress damages brain cells and impairs cognitive function, so selenium’s role in combating this stress is vital for maintaining cognitive health [129,206,207,208,209,210]. Selenium, contributing to antioxidant defenses, can reduce the risk of diseases by boosting immune function. It has been found that higher levels of selenium in the blood are associated with a reduced risk of cognitive decline and certain types of cancer [211,212]. Selenium is also crucial for the proper functioning of the thyroid gland and plays a vital role in immune health.

Zinc, essential for growth, development, and immune system function [213], is beneficial for the nervous system and may help prevent depression due to its tranquilizing properties. However, zinc deficiency can lead to a decline in cognitive functions and memory [214,215]. Zinc is also vital for the functionality of the antioxidant enzyme superoxide dismutase, which helps eliminate free radicals [216,217,218].

Copper, involved in energy production, connective tissue formation, and the integrity of the cardiovascular and immune systems, requires a balanced intake with zinc [219]. Copper is found in diverse food sources and is also available as a dietary supplement. While copper performs many vital biological functions, the relationship between its intake and the development of diseases like Alzheimer’s disease remains complex, with research suggesting that both excess and deficiency may have adverse consequences [220,221].

In summary, minerals such as magnesium, iron, selenium, zinc, and copper are integral to maintaining cognitive health, particularly in the aging population. Their supplementation can support various brain functions, from cognitive performance to reducing the risk of neurodegenerative diseases. This highlights the importance of ensuring adequate mineral intake, either through diet or supplementation, as part of a strategy to preserve cognitive function and overall brain health in older adults.

## 5. Practical Considerations

When considering supplementation with vitamins, minerals, antioxidants, and other dietary supplements for cognitive health, it’s crucial to take into account several individual factors. These include age, sex, nutritional status, lifestyle, stress levels, physical activity, season, dietary habits, and specific diets [222,223,224]. Customizing supplementation for each individual is essential for maximizing cognitive health benefits. As we age, our body’s ability to absorb nutrients decreases, and our nutritional needs change [225,226]. Older adults may need higher doses of certain vitamins like B12, vitamin D, and calcium. However, excessive intake of certain supplements can be harmful, so it is important to adjust dosages appropriately [156]. Men and women have different nutritional requirements [64]. For instance, women may need more iron due to menstruation, while men may require more zinc. Post-menopausal women often need increased calcium and vitamin D to maintain bone health. Nutritional status should be assessed to determine any existing deficiencies. For example, a person with low vitamin D levels might need a higher supplementation dose compared to someone with adequate levels. Factors like smoking or alcohol consumption can affect the body’s nutrient levels. Smokers may need more vitamin C, while heavy drinkers might require additional B vitamins. High stress can deplete certain nutrients faster, such as B vitamins and vitamin C. Individuals under significant stress might benefit from higher doses of these. Active individuals have higher metabolic rates and might require more of certain nutrients and antioxidants to combat increased oxidative stress from physical exertion [227]. Vitamin D supplementation is particularly important in winter months or for individuals with limited sun exposure. In contrast, during summer, this might be less necessary. Vegetarians and vegans might lack certain nutrients like B12, iron, and omega-3 fatty acids, which are commonly found in animal products. People on specific diets might need to compensate for these gaps with supplements. It is important to consider personal health conditions and medications that can affect nutrient absorption. For instance, some medications can deplete certain vitamins, requiring supplementation.

Periodic blood tests can help monitor nutrient levels and adjust supplementation as needed. A diet rich in fruits, vegetables, whole grains, lean proteins, and healthy fats is preferred. Supplementation should not replace a healthy diet, but rather complement it. Before starting any supplement regimen, patients should consult with a healthcare provider, especially for those with health conditions or taking medications. In general, high doses of supplements should be avoided to avoid adverse effects. It should be emphasized that high-quality supplements from reputable sources should be preferred to ensure safety and efficacy. In conclusion, while Table 6 provides a comprehensive list of supplements beneficial for cognitive health, the approach to supplementation should be personalized, considering the unique needs and circumstances of each individual. This tailored approach ensures optimal benefits for brain health and cognitive function.

## 6. Conclusions

The comprehensive analysis presented in this paper underscores the profound significance of nutritional supplementation in the preservation of cognitive health, particularly in the context of aging. Vitamins, minerals, antioxidants, dietary polyphenols, carotenoids and omega-3 PUFAs emerge as pivotal elements that play varied and crucial roles in maintaining and enhancing cognitive function. A combination of these offer neuroprotective benefits, enhancing memory, attention, and overall brain function. Their potent antioxidant properties are instrumental in combating oxidative stress, a key factor in age-related cognitive decline. Vitamins B, C, D, and E each contribute uniquely to brain health, from supporting neurotransmitter synthesis to protecting against neuronal damage and supporting mood regulation. Minerals such as magnesium, iron, selenium, zinc, and copper are equally essential, each playing a role in various brain functions and protecting against cognitive decline. Magnesium’s role in neurotransmission, iron’s importance in oxygen transport, selenium’s antioxidant capabilities, zinc’s neuroprotective properties, and copper’s involvement in energy production highlight the intricate interplay of these nutrients and minerals in maintaining cognitive health. It is, however, imperative to emphasize the necessity of individualized supplementation strategies. Factors such as age, sex [228], nutritional status, lifestyle, stress levels, physical activity, and specific dietary habits must be considered when tailoring supplementation regimes. A one-size-fits-all approach is insufficient; rather, personalized plans based on individual needs and conditions are crucial for optimal cognitive health benefits. In future studies on dietary supplements, it will be crucial to assess not only the cognitive benefits, but also the effects on the aging process itself, including impacts on biological age [229,230], to comprehensively understand how these interventions may influence overall health and longevity [231,232,233,234].

In conclusion, the role of dietary supplements in maintaining and enhancing cognitive function cannot be overstated. While our understanding of the exact mechanisms continues to evolve, the current evidence strongly supports the inclusion of these nutrients in a balanced diet or as part of a targeted supplementation strategy. Future research, particularly randomized controlled trials, will be pivotal in further elucidating these relationships and refining recommendations for dietary supplementation to preserve cognitive health, especially in the aging population. As we advance our understanding, it becomes increasingly clear that a holistic approach, combining dietary, lifestyle, and supplementation strategies, is essential for maintaining cognitive health and quality of life in older adults.

In the pursuit of healthy longevity, the integration of dietary supplementation within a holistic prevention plan is paramount [235,236,237,238,239,240,241,242,243]. The importance of healthy nutrition [183] and adopting a proper lifestyle cannot be understated in the context of preserving mental health [244] and cognitive function, especially as we age [245]. However, it is crucial to recognize that supplementation alone is not a panacea; instead, it should be viewed as one component of a comprehensive approach to healthy aging. A balanced and nutritious diet, rich in vitamins, minerals, and essential nutrients, forms the foundation of this holistic approach. Incorporating diets such as the MIND diet and the Mediterranean diet [246,247,248], known for their emphasis on whole grains, fruits, vegetables, healthy fats, and lean proteins, can significantly contribute to cognitive health and overall well-being. Regular physical activity is another cornerstone of this holistic plan [249,250,251,252]. Exercise not only enhances physical health but also has been shown to improve cognitive function, reduce stress, and boost mood [54,253,254,255,256]. Cognitive training [257] and engaging in mental activities, such as board games, reading, and puzzles [258], is equally important in old age. These activities stimulate the brain, helping to maintain its function and slow cognitive decline. Lifestyle modifications, including the cessation of smoking and the maintenance of healthy sleep patterns [259], are critical. Smoking cessation can significantly reduce the risk of diseases that impair cognitive function, while adequate, quality sleep is essential for brain health and memory consolidation. Regular monitoring of vital health indicators like blood pressure [260,261,262], blood sugar [263,264], and cholesterol levels further contributes to preventing conditions that could adversely impact cognitive health. In prescribing a combination of dietary supplements and lifestyle changes, healthcare providers should consider the individual’s overall health, lifestyle, and nutritional needs [265,266,267]. This tailored approach ensures not just the mitigation of disease risk but also the enhancement of life quality and longevity. Emphasizing patient education about the importance of each component in this holistic plan is key [268,269]. Patients should understand how lifestyle choices and dietary habits, combined with appropriate supplementation, work synergistically to promote optimal health. The future of Healthy Longevity Medicine lies in this multifaceted approach. By embracing a holistic strategy that combines nutrition, dietary supplements, physical activity, mental engagement, and lifestyle modifications, we can significantly enhance the prospects of not only living longer, but also enjoying a healthier, more fulfilling life in our later years.

## Figures and Tables

**Figure 1 nutrients-15-05116-f001:**
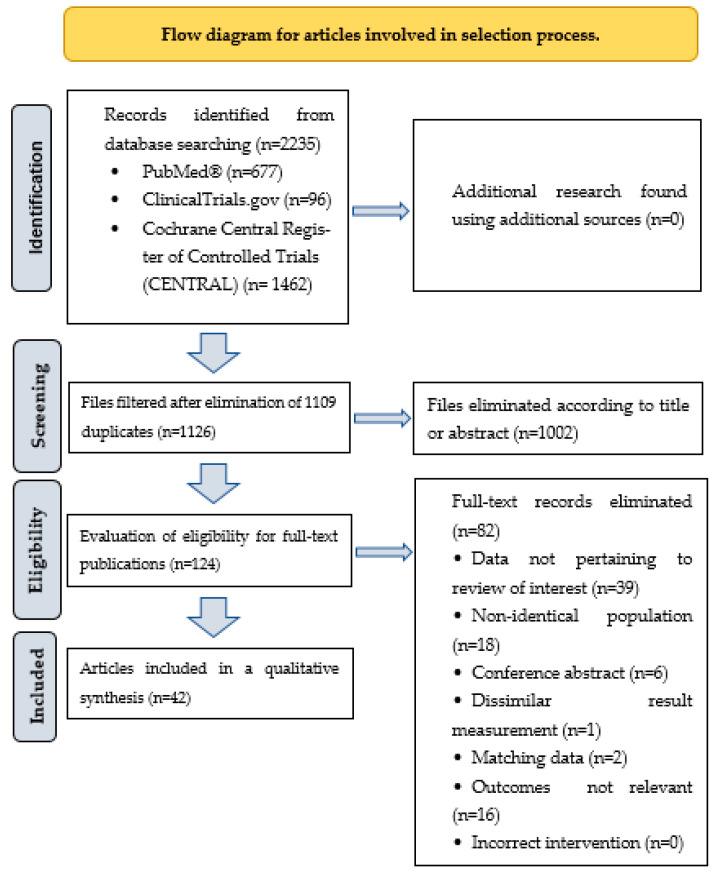
Flow diagram for articles involved in selection process.

**Table 1 nutrients-15-05116-t001:** Inclusion and exclusion criteria for studies included in the current review.

Inclusion Criteria	Description
Study design	Randomized controlled trial or human clinical trial.
Study population	Healthy people or patients admitted with a diagnosis of mild cognitive impairment or Alzheimer’s disease.
Intervention	Vitamins, antioxidants, minerals and micronutrients interventions.
Language of publication	No language restrictions applied.
Published articles	In the PubMed, ClinicalTrials.gov and Cochrane Central Register of Controlled Trials (CENTRAL) databases.
Output concepts	Different cognitive functions and their measurement tools, such as validated questionnaires: Full-Scale Intelligence Quotient (FSIQ), Wechsler Adult Intelligence Scale (WAIS), Mini Mental State Examination (MMSE), Stroop Color and Word Test (STROOP), Addenbrooke’s Cognitive Examination-Revised (ACE-R), Verbal Fluency Test. Cognitive index score and different cognitive function tests: attention, calculation, memory, verbal fluency, psychomotor speed, visual-constructional ability, neuropsychological function, reaction time, psychocognitive tests, etc.
**Exclusion Criteria**	
Animal experiments.	
In vitro studies.	
Macronutrients interventions, proteins, carbohydrates, fats, foods, medicines, pharmaceuticals, herbs, essential oils, melatonin and complex diets, such as Dietary Approach to Stop Hypertension (DASH diet), high-protein diet, ketogenic diet, low-fat diet, mediterranean diet, low glycaemic index (GI) diet, Mediterranean-DASH Intervention for Neurodegenerative Delay (MIND).	
Interventions for various diseases, such as: tumour diseases, epilepsy, post-traumatic stress disorder, anxiety, depression, stroke, multiple sclerosis, chronic cerebral ischaemia, polycystic kidney disease, opioid patients, psychosis, delirium, schizophrenia, bipolar disorder, frontal lobe atrophy, COVID-19 patients, sepsis, autism, attention deficit hyperactivity disorder (ADHD), Wernicke-Korsakoff syndrome, Fragile X syndrome, and Down syndrome.	
Interventions for different ages and conditions, such as: infancy, adolescence, pregnancy and interventions for athletes.	
Dietary advice, food and nutrition interventions.	
Short-term interventions (<4 weeks).	
Intravenous or intramuscular interventions.	
Dietary supplement interventions for underweight patients (body mass index (BMI): <18.5 kg/m^2^).	

**Table 2 nutrients-15-05116-t002:** Association between vitamin B supplementation and cognitive function.

Study	Design	Mean Follow-Up	Country	Sample Size	Average Age (Year)	Sex Male/Female (%)	Intervention	Main Results
Zhang C et al. [75]	RCT	30 days	China	102	41.0 ± 9.1	47/53	Magtein: 400 mg^®^ Vitamin D3: 80 IU Vitamin C: 12 mg Vitamin B6: 4 mg; Phosphatidylserine 50 mg. Total: 2 g/person/day.	Magtein^®^PS: Significant improvement in memory and cognition in healthy chinese adults (*p* < 0.001).
Ma F et al. [77]	RCT	6 months	China	240	70.5 ± 9.1	35/65	Four treatment groups: 800 µg FA only, 25 µg vitamin B12 only, FA, and vitamin B12 supplementation or control group.	Supplementation with FA and Vitamin B12: Significant Improvements in FSIQ (d = 0.169, *p* = 0.024), verbal IQ (d = 0.146, *p* = 0.033), Information (d = 0.172, *p* = 0.019), and Digit Span Scores.
Gibson GE et al. [73]	RCT	12 months	USA	70	75.7 ± 7.0	41.4/58.6	Benfotiamine treatment (300 mg/day twice a day) versus placebo group.	Benfotiamine Group vs. Placebo Group: 43% lower increase in ADAS-Cog Scores and 77% less worsening in CDR (*p* = 0.034) in Benfotiamine Group.
Kwok T et al. [79]	RCT	24 months	Hong Kong	279	78.0 ± 5.3	56.1/43.9	MCI patients administered 500 μg methylcobalamin and 400 μg FA orally once daily.	Supplementation with Vitamin B12 and FA: No reduction in cognitive decline in older individuals with MCI and elevated serum homocysteine.
Bai D et al. [76]	RCT	6 months	China	138	68.3 ± 6.3	40/60	FA (60.0 mg/day) + DHA (8 mg/day), FA (800.0 mg/day), DHA (8 mg/day) versus placebo.	FA, DHA, and FA + DHA vs. placebo: improvements in FSIQ, arithmetic, and picture complement scores with folic acid; FSIQ, information, arithmetic, and digit span scores with DHA; greater enhancements in arithmetic (1.67, 95% CI 1.02 to 2.31) and digit span (1.33, 95% CI 0.24 to 2.43) scores with FA + DHA.
Ma F et al. [74]	RCT	24 months	China	180	74.8 ± 2.8	42.7/57.3	FA 400 µg/day.	FA supplementation linked to improved cognitive function and reduced blood levels of Aβ-related biomarkers in MCI (*p* < 0.05).
Li M et al. [78]	RCT	6 months	China	240	70.4 ± 6.7	42.5/57.5	Intervention groups: FA + DHA (FA 800 μg/d + DHA 800 mg/d), FA (FA 800 μg/d), DHA (DHA 800 mg/d), and placebo.	Daily oral FA, DHA, and combined use for 6 months: significant improvements in FSIQ and select WAIS subtests compared to placebo (*p* < 0.05).

ADAS-Cog: Alzheimer’s Disease Assessment Scale-Cognitive Subscale; CDR: clinical dementia rating; DHA: docosahexaenoic acid; FA: folic acid; FSIQ: full-scale intelligence quotient; IQ: intelligence quotient; IU: international unit; Magtein: Magnesium L-threonate; Magtein^®^PS: magnesium L-threonate (Magtein) and phosphatidylserine-based formula; MCI: mild cognitive impairment; RCT: randomized controlled trial; WAIS: Wechsler Adult Intelligence Scale.

**Table 3 nutrients-15-05116-t003:** Association between vitamin C and antioxidant supplementation and cognitive function.

Study	Design	Mean Follow-Up	Country	Sample Size	Average Age (Year)	Sex Male/Female (%)	Intervention	Main Results
Sim M et al. [82]	RCT	4 weeks	Republic of Korea	214	20–39	39.3/60.7	500 mg vitamin C twice a day.	Vitamin C supplementation notably increased attention and work absorption (*p* = 0.03), with a clear tendency towards fatigue improvement.
Morató X et al. [86]	RCT	12 months	Spain	50/50	73.1 ± 7.5	40/60	Standardized extract of Ginkgo biloba EGb 761 240 mg tablets were given orally.	No significant differences between groups in MMSE, CDR, NBACE scores, or amnestic profile; higher scores in irritability/lability parameter (*p* = 0.006) and BDS (*p* = 0.048) in control group.
Thaung Zaw JJ et al. [80]	RCT	24 months	Australia	125	45–85	100% female	75 mg trans-resveratrol or placebo per day.	Resveratrol supplementation led to 33% improvement in overall cognitive performance (Cohen’s d = 0.170, *p* = 0.005).
Lai S et al. [84]	RCT	6 months	Italy	59	45.1 ± 10.7	55.9/44.1	ALA 1.6 g/day.	BDI-II, HAM-D, MMSE tests showed significant improvement (*p* = 0.007, *p* < 0.001, *p* < 0.001) in patients treated with ALA compared to control group.
Foroumandi E et al. [81]	RCT	4 months	Iran	82	72.0 ± 2.5	34.1/65.9	Intervention group patients received 500 mg dry extract (5 cc) of fenugreek seed extract.	Positive effects on memory (*p* < 0.001), quality of life (*p* < 0.001) and selective oxidative index level.
Baker LD et al. [87]	RCT	3 years	USA	2262	73.0	40/60	Daily administration of cocoa extract (containing 500 mg/day flavanols) versus placebo.	Cocoa extract had no effect on global cognition (mean z-score = 0.03, 95% CI: −0.02–0.08; *p* = 0.28).
Bell L et al. [88]	RCT	12 weeks	UK	60	18–30	10/90	400 mg of GSPE extract daily.	400 mg of GSPE did not consistently improve cognitive function in healthy young adults.
Stringham NT et al. [83]	RCT	6 months	Greece	59	18–25	45.7/54.3	Carotenoids lutein and zeaxanthin, along with the zeaxanthin isomer meso-zeaxanthin (13 mg/day or 27 mg/day total).	For cognitive measures, all scores for composite memory, verbal memory, sustained attention, psychomotor speed, and processing speed improved significantly in treatment groups (*p* < 0.05 for all) and remained unchanged in the placebo group.
Yoon J et al. [85]	RCT	12 weeks	Japan	36 + 36	69.5 ± 7.6	47/53	Participants received 3 g of DOTP or placebo in olive oil twice daily for 12 weeks.	Among cognitive domains, complex attention had a significant time × group interaction effect (*p* = 0.049) between the DOTP and placebo groups. Time effects were significant (*p* < 0.05) for psychomotor speed, reaction time, cognitive flexibility, processing speed, and executive function domains.
Hashimoto M et al. [89]	RCT	12 months	Japan	44	70.2 ± 1.4	52.4/47.6	Randomized participants in the PO group received soft gelatin capsules containing 1.47 mL of PO daily, and those in the PO + POPP group received soft gelatin capsules containing both 1.47 mL of PO and 1.12 g PP daily.	At the end of intervention, the POPP group showed significantly higher cognitive index scores than the PO group, POOP may improve age-related cognitive impairment in healthy elderly people.

ALA: alpha lipoic acid; BDI-II: Beck Depression Inventory-II; BDS: Blessed Dementia Scale; CDR: clinical dementia rating; DOTP: desert olive tree pearl; GSPE: grape seed polyphenol; HAM-D: Hamilton Depression Rating Scale; MMSE: Mini-Mental State Examination; PO: Perilla frutescens seed oil; POPP: PO + ponkan powder.

**Table 4 nutrients-15-05116-t004:** Association between vitamin D supplementation and cognitive function.

Study	Design	Mean Follow-Up	Country	Sample Size	Average Age (Year)	Sex Male/Female (%)	Intervention	Main Results
Jia J et al. [93]	RCT	12 months	China	210	68.0 ± 5.9	45/55	Patients received 800 IU/day of vitamin D.	The FSIQ and cognitive test score were significantly higher in the intervention group than in the control group (*p* < 0.001).
Bischoff-Ferrari HA et al. [92]	RCT	3 years	Switzerland	2157	74.9	38.3/61.7	2000 IU/day of vitamin D3, 1 g/day of omega-3 strength-training exercise program.	Among adults aged 70 years or older, treatment with vitamin D3, omega-3, or a strength-training exercise program did not result in statistically significant differences in improving cognitive function.
Zajac IT et al. [90]	RCT	6 months	Australia	436	60–90	48.4/51.6	600 IU/day of vitamin D3	No effect.
Yang T et al. [94]	RCT	12 months	China	183	67.2 ± 6.1	46/54	800 IU/day of vitamin D3	The ANOVA showed improvements in the FSIQ, information, digit span, vocabulary, block design, and picture arrangement scores in the vitamin D group over the placebo group (*p* < 0.001).
Byrn MA et al. [99]	RCT	12 weeks	USA	206	55.71	17/83	Administration of either weekly vitamin D3 supplementation (50,000 IU) or 5000 IU cholecalciferol once a week.	No significant differences in cognitive outcomes between participants who received high-dose therapy and those who received low dose.
Castle M et al. [95]	RCT	1 year	USA	138	58 ± 6	100% female	Vitamin D3 supplementation (600, 2000, or 4000 IU/day).	The CANTAB test results indicated that the 2000 IU/d group, when compared to other groups, performed better in PAL test parameters (*p* < 0.05). RTI was slower in the 4000 IU/d compared to 600 IU/d group for the 5-choice test (*p* < 0.01).
Jorde R et al. [91]	RCT	4 months	Norway	422	52	52.9/47.1	Vitamin D 100,000 IU administered as a bolus dose followed by 20,000 IU per week versus placebo.	Vitamin D supplementation did not improve cognitive function during a four-month intervention.
Ghaderi A et al. [96]	RCT	24 weeks	Iran	64	59.2 ± 11.3	53/47	Administration of either 50,000 IU vitamin D supplements (*n* = 32) or placebo (*n* = 32) every 2 weeks.	Subjects who were administered vitamin D had a significant reduction in IGT (β −6.25; 95% CI, −8.60 to −3.90; *p* < 0.001), and significant increases in VFT (β 2.82; 95% CI, 0.78–4.86; *p* = 0.007), immediate LM (β 1. 32; 95% CI, 0.27–2.37; *p* = 0.01), reverse DGS (β 2.06; 95% CI, 1.18–2.94; *p* < 0.001) and VWM (β 0.75; 95% CI, 0.33–1.16; *p* = 0.001).
Schietzel S et al. [98]	RCT	2 years	Switzerland	273	70.3	46.5/53.5	2000 or 800 IU vitamin D3/day	No effect.
Beauchet O et al. [100]	RCT	3 months	Canada	40	≥65	100% female	Fortified yogurt (400 IU vitamin D and 800 mg calcium).	Fortified yogurts with vitamin D and calcium maintained global cognitive performance (MMSE score (*p* = 0.022). Global cognitive performance decreased in the control group.
Owusu JE et al. [101]	RCT	3 years	USA	260	68.2 ± 4.9	100% female	Adminsitration of vitamin D (adjusted to achieve a serum level > 30 ng/mL) with calcium (diet and supplement total of 1.200 mg)	There is no evidence that vitamin D intakes above the recommended daily allowance are needed to prevent cognitive decline in this population.
Hu J et al. [97]	RCT	12 months	China	181	67.22 ± 6.1	44.5/55.5	Administration of 800 IU/day of vitamin D.	The mean scores of information, DGS, vocabulary, block design and picture arrangement tests in the vitamin D3 group were significantly higher than that in the placebo group both before and after adjustment. In addition, the performance of FIQ (*p* < 0.001, d = 0.70), VIQ (*p* < 0.001, d = 0.77) and PIQ (*p* < 0.001, d = 0.70) was consistent with the five subtests mentioned above.
Macpherson H et al. [102]	RCT	6 months	Australia	147	70.2 ± 6.1	30/70	Daily vitamin D (1000 IU) and omega-3 (900 mg EPA, 600 mg DHA) and protein (20 g) supplementation.	There were no significant between-group differences in cognition at 6 or 12 months.

DHA: docosahexaenoic acid; EPA: Eicosapentaenoic acid; FSIQ: full scale intelligence quotient; PAL: Paired Associates Learning; RTI: reaction time; IGT: Iowa Gambling Task; VFT: Verbal Fluency Test; LM: Logic Memory; DGS: Digit Span; VWM: visual working memory; FIQ: full IQ; VIQ: verbal IQ; PIQ: performance IQ.

**Table 5 nutrients-15-05116-t005:** Association between omega-3 PUFA supplementation and cognitive function.

Study	Design	Mean Follow-Up	Country	Sample Size	Average Age (Year)	Sex Male/Female (%)	Intervention	Main Results
Lin PY et al. [108]	RCT	24 months	Taiwan	163	77.8 ± 8.4	66.2/33.8	163 patients were randomly assigned to DHA (0.7 g/day), EPA (1.6 g/day), or EPA (0.8 g/day) + DHA (0.35 g/day) group for 24 months.	A statistically significant difference in cognitive, functional, and mood status scores, biochemical profiles, and inflammatory cytokines levels was not determined between the placebo and treatment groups.
Nolan JM et al. [103]	RCT	12 months	Ireland	50 + 27	≥65	53/47	Patients consumed 1 g fish oil (of which 500 mg DHA, 150 mg EPA), 22 mg carotenoids (10 mg lutein, 10 mg meso-zeaxanthin, 2 mg zeaxanthin), and 15 mg vitamin E daily.	The active group performed better in objective measures of AD severity (i.e., memory and mood), with a statistically significant difference in the clinical collateral for memory (*p* < 0.001).
Giudici KV et al. [109]	RCT	3 years	France	1445	75.3 ± 4.4	35.8/64.2	ω-3 (800 mg DHA and 225 mg EPA/day)	No effect.
Arellanes IC et al. [113]	RCT	6 months	USA	33	68 (58–90)	18.2/81.8	33 individuals were provided with a vitamin B complex (1 mg of vitamin B12, 100 mg of vitamin B6 and 800 mcg of folic acid per day) and randomized to 2152 mg of DHA per day or placebo over 6 months.	No effect.
Rasmussen J. [107]	RCT	24 months	Finland, Germany, Netherlands, Sweden	311	>65	45/55	Patients were given a combination of DHA, EPA, uridine monophosphate, choline, phospholipids, selenium, folic acid, and vitamins B12 and E (>200% the recommended daily intake).	This intervention had the potential to improve the progression of Alzheimer’s disease (attention, memory, executive function *p* < 0.05).
Stavrinou PS et al. [104]	RCT	6 months	Cyprus	36	78.8 ± 7.3	38.8/61.2	20 mL dose of a formula containing a mixture of omega-3 (810 mg EPA and 4140 mg DHA) and omega-6 FA (1800 mg gamma-LA and 3150 mg LA) (1:1 *w*/*w*), with 0.6 mg of vitamin A, vitamin E (22 mg) plus pure γ-tocopherol (760 mg).	A significant interaction between supplementation and time was found on cognitive function (MMSE, ACE-R, STROOP) functional capacity (6 min walk test; *p* = 0.028) fatigue (*p* < 0.001), physical health (*p* = 0.007), and daily sleepiness (*p* = 0.007).
Mengelberg A et al. [110]	RCT	12 months	New Zealand	30 + 30	72.3 ± 6.1	69.8/30.2	1491 mg of DHA + 351 mg of EPA per day.	No effect.
Atmadja T et al. [105]	RCT	90 days	Indonesia	29	66.1 ± 5.3	20.6/79.4	Subjects were divided into three groups: SO, CFO, and CO with omega-3 (catfish oil enriched with omega-3). The intervention involved 1000 mg of oil/day.	Significant effects on oxidative stress and cognitive function (*p* < 0.05), and significantly increased MMSE score (*p* < 0.05).
Leckie RL et al. [111]	RCT	18 weeks	USA	271	30–54	43.5/56.5	Fish oil capsules (1400 mg/day of EPA and DHA).	No effect.
Kuszewski JC et al. [114]	RCT	16 weeks	Australia	64	65.8 ± 1.4	44/56	Randomly assigned to either fish oil (2000 mg/d of DHA + 400 mg/d of EPA), curcumin (160 mg/d), or a combination.	Fish oil improved CVR to a processing speed test (4.4% ± 1.9% vs. −2.2% ± 2.1%; *p* = 0.023) and processing speed in males only (Z-score: 0.6 ± 0.2 vs. 0.1 ± 0.2; *p* = 0.043).
Patan MJ et al. [106]	RCT	26 h	UK	310	25–49	35/65	Participants consumed either 900 mg of DHA/d and 270 mg of EPA/d (DHA-rich oil), 360 mg of DHA/d and 900 mg of EPA/d (EPA-rich oil), or 3000 mg/d of refined olive oil (placebo).	EPA supplementation improved global cognitive function, superior to oil enriched with DHA; improved memory accuracy compared with DHA (*p* = 0.034)
Sueyasu T et al. [112]	RCT	24 weeks	Japan	71	65.7 ± 1.0	45.1/54.9	PUFA in Combination with LZ (containing 120 mg ARA, 300 mg DHA, and 100 mg EPA per day) combined with LZ (containing 10 mg lutein and 2 mg zeaxanthin per day).	LCPUFAs + LZ supplementation did not significantly affect memory function.

ACE-R: Addenbrooke’s Cognitive Examination–Revised; CFO: commercial fish oil; CO: catfish oil; CVR: cerebrovascular responsiveness; DHA: docosahexaenoic acid; DHA: docosahexaenoic acid; EPA: Eicosapentaenoic acid; FA: fatty acid; LA: linolenic acid; LCPUFAs + LZ: Long-Chain Polyunsaturated Fatty Acids in Combination with Lutein and Zeaxanthin; MMSE: Mini-Mental State Examination; PUFA: Polyunsaturated Fatty Acid; SO: soybean oil; STROOP: Stroop Color and Word Test.

**Table 6 nutrients-15-05116-t006:** Vitamins, minerals, antioxidants and other dietary supplements that can help support cognitive functions.

Vitamins	Minerals	Antioxidants
Vitamin B Complex	Magnesium	Beta-carotene
Vitamin A	Selenium	Lutein
Vitamin C	Copper	Lycopene
Vitamin D	Iron	Coenzyme Q10
Vitamin E	Zinc	Polyphenols
Vitamin K	Potassium	Curcumin (Turmeric)
Choline	Calcium	Acetyl-L-Carnitine
Omega 3 Polyunsaturated Fatty Acids (fish oils)		
Probiotics, Prebiotics, Synbiotics		

Source: own ed. This review does not offer a detailed description of the effects of probiotics, prebiotics, synbiotics, carnitine, coenzyme Q10 supplementation.

## Data Availability

Data sharing is not applicable to this article as no new data were created or analyzed in this study.

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
