# Peer review of "Improving Cognitive Function with Nutritional Supplements in Aging: A Comprehensive Narrative Review of Clinical Studies Investigating the Effects of Vitamins, Minerals, Antioxidants, and Other Dietary Supplements"

_nutrients, 2023, doi:10.3390/nu15245116_

Round 1

Reviewer 1 Report

Comments and Suggestions for Authors

nutrients-2744616-review

The Role of Nutritional Supplements on Cognitive Function, or Which Vitamins, Minerals, Antioxidants and Other Dietary Supplements Can Improve Cognitive Performance—A State-of-the-Art Review

This is very extensive review describing existing data but lacking conclusions on what the optimal supplementation should be at the state-of-the-art knowledge.

The statement in the abstract “Our results suggest that there is a strong positive association between the supplementation of various dietary supplements, vitamins, minerals, antioxidants, and polyunsaturated fatty acids and cognitive function.” is rather vague and inconclusive.

Specific comments

The Authors describe existing studies on vitamins B, C, D, E, K, omega-3 fatty acids, general antioxidants and some minerals, usually concluding that the existing data are “confirmed by our resent review”.

Therefore, after reading this review, one is confirmed that multiple studies examined the role of different supplements on cognitive function but does not know which of those substances can improve cognitive performance and should be supplemented to the patients.

Comments on the Quality of English Language

-

Author Response

Answer-1

Thank you for your valuable constructive feedback. we have revised the manuscript and am attaching it for your review. The optimal supplementation of vitamins and minerals depends on factors such as age, gender, nutritional status, lifestyle, stress levels, physical activity, dietary habits, and specific diets, such as those involving omega-3. It also varies with the seasons, for example, in the case of vitamin D. Both vitamins and dietary supplements depend on an individual's current needs and health conditions, including medications taken, etc.

It is always necessary to tailor the supplementation of vitamins and minerals to the individual, and if needed, monitor serum levels because their deficiency can cause severe harm to the individual, as in the case of vitamin D hypovitaminosis. There is no one-size-fits-all optimal supplement for everyone. Vitamins, minerals, antioxidants, omega-3, etc., as dietary supplements, are worth considering and applying periodically to ensure optimal health of cognitive functions and the immune system, as we may not always meet these needs through our diet, for example, with vitamin D and omega-3.

Furthermore, the content of vitamins and minerals in our food has significantly decreased in recent years, as supported by various international studies, revealing unfortunate facts such as the reduction in C-vitamin, calcium, folic acid, magnesium, etc. The reasons are complex, but intensive, large-scale agriculture, and the use of the latest yield-enhancers, such as fertilizers and pesticides, actively reduce the nutrient composition of the soil and vegetables and fruits.

Source:

  1. Drewnowski A, Rehm CD. Vegetable cost metrics show that potatoes and beans provide most nutrients per penny. PLoS One. 2013;8(5):e63277. Published 2013 May 15. doi:10.1371/journal.pone.0063277
  2. https://www.scientificamerican.com/article/soil-depletion-and-nutrition-loss/

I sincerely appreciate your valuable review.

Best regards,

Mónika Fekete, MD, PhD

Reviewer 2 Report

Comments and Suggestions for Authors

Review No. 2744616

The amount of work and the amount of research analyzed should be appreciated. Unfortunately, perhaps due to too extensive material, the work was lost. The way of presenting the results, regardless of the tabular or descriptive version, is superficial and quite monotonous, i.e.: description of the supplementation (exhaustive or not exhaustive) and description of the effects. Only tests for examining neurological functions and whether or not statistics are variable.

The tables should be refined and the information they contain should be analyzed and filtered again. Below are some examples that illustrate this point*.

According to the reviewer, studies in which the dosage schedule and dose levels of individual substances are missing or the quantitative and qualitative composition of the supplement used are unknown, such as  TwendeeX  or  Biokesum (Table 2)  should be excluded.

Generally, the number of presented studies should be narrowed down and focused on their more detailed analysis.

 Introduction

Supplement the Introduction with a justification for including probiotic supplementation in the analysis.

Results

Lines 273 - 274: the sentence should be reworded: if it is not a problem to take 4000 IU of vitamin D, why should you avoid taking it in the range of 500 - 2000 IU?

Discussion

Vitamin D

Lines 468– 470: 2000 IU is the RDA for vitamin D not only in Hungary. The reviewer refers to the work published in Nutrients (2023); 15, 695. http://doi.org/103390/nu15030695.

The Limitation subsection should rather be titled: Summary.

Conclusions

The role of probiotics and minerals was omitted in the Conclusions, but coenzyme Q10 was included.

*Examples of entries in Tables with reviewer reservations:

Table 2

1.       Double entries in the Sex Male/Female column are incomprehensible (e.g. Table 2, Morato X et al. (67) given: 40/60 and 36/64).

2.       Assmann KE (70) - incomplete information regarding supplementation: the vitamin composition is provided and the influence of PUFA and MUFA is discussed.

3.       Foroumandi et al. (71): 5cc units need to be explained.

4.       Crosta F et al. (83): supplement B12 dose.

Table 3

1.       Jia et al. (84): The description of the results should be specified - what does may improve mean?

2.       Byrn M.A. et al. (88): what does matching comparator mean?

3.       Castle M et al. (90): what does differential effects mean? Such a description does not provide any information.

4.       Beauchet O. (94): the result of these studies is presented too generally and does not contribute anything (no comparison, no statistical analysis)

5.        Hu J et al. (96): what does significantly improved mean?

Table 4

1.        Rasmussen J (105): no dosage regimen.

2.       Assmann KE et al. (70): the results are inconsistent with the type of supplementation (see note to Table 2).

3.       Sueyasu T et al. (113): no dose sizes.

General remarks to Tables:

The layout of the columns should be corrected so that the text in each of them is legible and the headings should be completed (Sex Male/Female %).

The method of applying and using abbreviations in the text should be sorted out: in the current version of the work, the abbreviations are explained under the Tables and/or in the content of the Tables.

Content not directly related to the effect of supplementation on neurological functions, such as bone resorption (Table 4, Ichinese T. et al. (104)), should be removed, and conclusions should be removed (Ghahremani M. et al. (98), Table 3) and also certainly studies in which the route of administration excludes supplementation (vitamin B12 administered i.m., Table 1, Zhou L. et al. (58)).

Review summary

Authors should select the material, e.g. by adopting the following criteria: - reports from the last 3 years - selection of works dealing exclusively with the impact of supplementation on neurological functions; - RCT studies with statistical analysis; - studies in which the dosing regimen, dose levels and results were clearly described. They should present the results in one version, e.g. tabular, and then discuss them in the Discussion in the context of the dose and composition. They should also, based on research conclusions, suggest the most beneficial qualitative and quantitative compositions of supplements that should be taken  potentially effectively to improve brain function.

The material in the current version is too extensive, to the detriment of the reader. The work should surprise  with something. Meanwhile,  most of the Discussion is textbook information about most of the ingredients taken in the form of supplements, i.e. vitamins and minerals

Author Response

Thank you for your valuable constructive feedback. We have revised the manuscript and the tables, and I am attaching them for your review. This review spans the past 5 years as we aimed to provide more comprehensive data than the last 3 years. The optimal supplementation of vitamins and minerals depends on factors such as age, gender, nutritional status, lifestyle, stress levels, physical activity, dietary habits, and the type of diet employed (Mediterranean, DASH, MIND, etc.), for example, in the case of omega-3. It also varies with the seasons (summer or winter), as in the case of vitamin D. Both for vitamins and dietary supplements, it depends on the individual's current needs and health conditions, including medications taken, etc. Precise dosages cannot be recommended because it is always necessary to tailor the supplementation of vitamins and minerals to the individual. If needed, monitoring serum levels is crucial, as deficiency can cause severe harm to the patient, as in the case of vitamin D hypovitaminosis. Unfortunately, there is no one-size-fits-all optimal vitamin supplementation. Vitamins, minerals, antioxidants, omega-3, etc., as dietary supplements, are worth considering and applying periodically to ensure optimal health of cognitive functions and the immune system, as our diets may not always cover these needs, for example, with vitamin D and omega-3. Moreover, the vitamin and mineral content of our foods has significantly decreased in recent years, as confirmed by several international studies. The reasons are complex, but intensive, large-scale agriculture and the use of the latest yield enhancers such as fertilizers and pesticides actively reduce the nutrient composition of the soil and vegetables and fruits.

Source:

  1. Drewnowski A, Rehm CD. Vegetable cost metrics show that potatoes and beans provide most nutrients per penny. PLoS One. 2013;8(5):e63277. Published 2013 May 15. doi:10.1371/journal.pone.0063277
  2. https://www.scientificamerican.com/article/soil-depletion-and-nutrition-loss/

Our MESH search terms did not include the following in the methodology: probiotics, prebiotics, symbiotics because this would expand the current database of this article by approximately 50-70 recently published articles. We plan to include these keywords in a new review article to be published in Nutrients.

Missing values in the tables have been filled in and clarified as requested by the Assessor. The requested research has been removed from the manuscript: Zhou L et al (58), Tadokoro K et al (64), Ton AMM et al (65), Tamtaji OR et al (66), Lau H et al (79), Ghahremani M et al (98) Kamalashiran C et al (63).

Thank you very much for this valuable review,

Best regards,

Mónika Fekete MD, PhD

Reviewer 3 Report

Comments and Suggestions for Authors

Authors reviewed “The Role of Nutritional Supplements on Cognitive Function, or ………………………… Can Improve Cognitive Performance—A State-of-4 the-Art Review" by Fekete et al. The write up is very clear and written precisely. However, following concerns need to be addressed and should be improved more.

Authors should add the inclusion and exclusion criteria as well in flow diagram or make a separate flow diagram of the selection criteria.

In flow diagram the text is not readable, Identification screening etc., authors should correct it.

In methods section authors should write the full form of MESH.

Authors should report how long patients have been taking nutritional supplements based on the literature reviewed 

Authors should include at least one figure/diagram showing various nutritional supplements that improve cognitive abilities.

In table 1, Ma F et al. (54), authors should report the percentage sex ratio and dosage amount.

In table 1, Gibson GE et al. (55), Write the dosage quantity of benfotiamine and placebo

Thanks

Author Response

Thank you very much for your valuable comments. We have created a table outlining the inclusion and exclusion criteria and incorporated it into the manuscript. The letters in the process diagram have been enlarged, improving readability for the reader. I appreciate this suggestion. In the methodology, we utilized MESH terms and have corrected them in response to your feedback—thank you for pointing that out. Patients in this review took vitamins, antioxidants, and other dietary supplements in a highly diverse manner, with varying durations in each study. Some took them for weeks, while others took them for months. The tables now include these durations. We have also prepared a new table presenting various dietary supplements that enhance cognitive abilities for the reader, and this has been included in the manuscript. Missing values in the tables have been filled in and clarified. Thank you very much. I sincerely appreciate your valuable critique.

Best regards,

Mónika Fekete MD, PhD

Round 2

Reviewer 1 Report

Comments and Suggestions for Authors

-

Comments on the Quality of English Language

-

Author Response

Thank you very much for your work!

With kind regards,

Mónika Fekete and the Co-authors

Reviewer 2 Report

Comments and Suggestions for Authors

1.        The Tables still contain results that are not supported by statistics as given below:

Table 2

Wu Y et al. (78)

Table 3

Assmann KE et al. (85)

Nakamura Y et al. (88)

Wsakurai K et al. (89)

Shiojima Y et al. (90)

Choi WY et al. (92)               

Hashimoto M et al. (93)

Crosta F et al. (97)

Table 4

Kang JH et al. (103)

Table 4 (should be 5):

Power R et al. (112)

Ichinose T et al. (117)

Ogawa T et al. (125).

2.        Table 3 – the abbreviation TwX in Table 3 is not applicable.

3.        The method of using abbreviation should be standardized, i.e. an abbreviation in the Table and an explanation under the Table.

4.         Table 4: Association between omega-3PUFA supplementation and cognitive function should be Table 5, with the change of references in the text.

5.        If the Authors develop practical considerations based on the analysed reports, they should do it consistently, i.e. choose the type of supplements/compounds, analyse the results and draw conclusions or refrain from discussing individual group of supplements. The note applies to probiotics, prebiotics, synbiotics, carnitine and coenzyme Q10.

Author Response

Thank you for your valuable critique and work.

We have removed the requested publications from the tables and consistently represented abbreviations in both tables, the changes are highlighted by red. Vitamin D was administered together with omega-3 (see Table 4), so we would like to include it among vitamin D supplements.

We have removed from the manuscript: Wu Y et al. (78), Power R et al. (112), Ichinose T et al. (117), Ogawa T et al. (125), Kang JH et al. (103), Assmann KE et al. (85), Nakamura Y et al. (88), Sakurai K et al. (89) Shiojima Y et al. (90) Choi WY et al. (92) Hashimoto M et al. (93) Crosta F et al. (97), and the abbreviation "TwX."

We have also removed statements regarding probiotics, prebiotics, synbiotics, carnitine, and Q10 coenzyme.

Thank you very much for your work.

With kind regards,

Mónika Fekete and the Co-authors